# Incidence, Severity and Clinical Factors Associated with Hypotension in Patients Admitted to an Intensive Care Unit: A Prospective Observational Study

**DOI:** 10.3390/jcm11226832

**Published:** 2022-11-18

**Authors:** Lotte E. Terwindt, Jaap Schuurmans, Björn J. P. van der Ster, Carin A. G. C. L. Wensing, Marijn P. Mulder, Marije Wijnberge, Thomas G. V. Cherpanath, Wim K. Lagrand, Alain A. Karlas, Mark H. Verlinde, Markus W. Hollmann, Bart F. Geerts, Denise P. Veelo, Alexander P. J. Vlaar

**Affiliations:** 1Department of Anesthesiology, Amsterdam UMC, Location Academic Medical Center, University of Amsterdam, Meibergdreef 9, P.O. Box 22660, 1105 AZ Amsterdam, The Netherlands; 2Department of Intensive Care, Amsterdam UMC, Location Academic Medical Center, University of Amsterdam, Meibergdreef 9, P.O. Box 22660, 1105 AZ Amsterdam, The Netherlands; 3Cardiovascular and Respiratory Physiology Group, Technical Medical Center, University of Twente, Drienerlolaan 5, 7522 NB Enschede, The Netherlands; 4Laboratory of Experimental Intensive Care and Anesthesiology, Amsterdam UMC, Location Academic Medical Center, University of Amsterdam, Meibergdreef 9, P.O. Box 22660, 1105 AZ Amsterdam, The Netherlands

**Keywords:** area under the threshold, arterial waveform, blood pressure, cardiovascular, hemodynamics, hypotension, monitoring

## Abstract

Background: The majority of patients admitted to the intensive care unit (ICU) experience severe hypotension which is associated with increased morbidity and mortality. At present, prospective studies examining the incidence and severity of hypotension using continuous waveforms are missing. Methods: This study is a prospective observational cohort study in a mixed surgical and non-surgical ICU population. All patients over 18 years were included and continuous arterial pressure waveforms data were collected. Mean arterial pressure (MAP) below 65 mmHg for at least 10 s was defined as hypotension and a MAP below 45 mmHg as severe hypotension. The primary outcome was the incidence of hypotension. Secondary outcomes were the severity of hypotension expressed in time-weighted average (TWA), factors associated with hypotension, the number and duration of hypotensive events. Results: 499 patients were included. The incidence of hypotension (MAP < 65 mmHg) was 75% (376 out of 499) and 9% (46 out of 499) experienced severe hypotension. Median TWA was 0.3 mmHg [0–1.0]. Associated clinical factors were age, male sex, BMI and cardiogenic shock. There were 5 (1–12) events per patients with a median of 52 min (5–170). Conclusions: In a mixed surgical and non-surgical ICU population the incidence of hypotension is remarkably high.

## 1. Introduction

Annually, more than five million patients in the US are admitted to the intensive care unit (ICU) due to life-threatening conditions. The majority of these patients develop severe hemodynamic instability [1]. At present, prospective studies examining the incidence and severity of hypotension, in light of specific blood pressure (BP) thresholds in general ICU patients, are scarce [2,3]. The severity and duration of hypotensive events are independently associated with an increased risk of acute kidney injury, myocardial injury, prolonged hospital stay and mortality [3,4,5,6,7].

To date, most studies have focused on hypotension in patients with septic shock or, more specific, on the associations of different mean arterial pressure (MAP) thresholds with adverse outcomes in sepsis and surgical ICU patients [3,7,8,9]. Current ICU guidelines, mainly based on retrospective cohort studies, recommend at least a MAP of 65 mmHg to improve microcirculation and clinical outcomes [3,10]. Furthermore, these studies use absolute cut off values that do not take the severity and duration of hypotension into account. An alternative to the use of absolute MAP thresholds, is the use of time-weighted average (TWA), for which the relationship of severity to duration of hypotension is calculated [3]. In recent years, TWA has become a well appreciated variable to describe hypotension [3].

In this prospective observational study, the primary aim was to determine the incidence of hypotension in a mixed surgical and non-surgical ICU population. Secondary outcomes were the severity of hypotension expressed in TWA, the number of hypotensive events, duration of hypotensive events and clinical factors associated with hypotension. 

## 2. Materials and Methods

### 2.1. Study Design 

This prospective, observational, single center study was approved by the Medical Ethical Committee of Amsterdam UMC location AMC, the Netherlands in May 2018 (Source ID: W18_142#18.176). This study was registered with the Netherlands Trial Register (NTR7349). Data collection took place in the ICU between 9 September 2018 and 30 May 2019. The study was conducted in compliance with the Declaration of Helsinki (Fortaleza, Portugal, 2013), the Dutch Medical Research Involving Human Subjects Act (WMO, Geneva, Switzerland) and Good Clinical Practice (GCP, Geneva, Switzerland).

### 2.2. Participants

Patients ≥ 18 years, newly admitted to the ICU for an expected minimum of eight consecutive hours, either during the day or evening/night period and who received an arterial line as part of standard care, were eligible for inclusion. Trained and delegated members of the study team screened patients. Written informed consent was obtained from all participants. When consent could not be obtained prior to data collection, then consent was obtained afterwards. Exclusion criteria were: an inability to measure continuous BP data with an arterial line, controlled hypotension with a target MAP lower than 65 mmHg and logistic difficulties (i.e., transfer to another hospital).

### 2.3. Measurements

Continuous BP data were collected for seven to eight hours, with a CE-certified unblinded hemodynamic monitor (the Flotrac EV1000, Edwards Lifesciences LTD, Irvine, CA, USA) using the arterial line systems of patients. Intra-arterial BP was measured with a five French cannula in the radial artery or in some cases the brachial or femoral artery. Measurements were displayed in mmHg with a sample frequency of 100 Hz and hemodynamic variables were averaged every 20 s (0.05 Hz). The Flotrac EV1000 accurately detects rapid changes in BP [11]. The device is used in operating theatres and in the ICU to monitor hemodynamic variables derived directly from arterial waveform analyses. Displayed variables include MAP, systolic arterial BP, diastolic arterial BP, heart rate (HR) and those derived from the arterial waveform such as cardiac output (CO), cardiac index (CI), stroke volume (SV), SV index (SVI), stroke volume variation (SVV) and pulse pressure variation (PPV). Pressure waveforms and hemodynamic data are stored on an internal disk. The de-identified data files were copied to a USB flash drive, were stored on a secure hospital server and were only accessible to members of the study team.

### 2.4. Analyses

Hypotension was defined as an absolute threshold MAP value of <65 mmHg, based on literature, ICU guidelines and the international consensus statement [2,3,6,8]. A single hypotensive event was defined as a MAP below 65 mmHg, for at least ten seconds, and with at least one minute in-between single hypotensive events. Due to the lack of a gold standard regarding accurate continuous prospective BP measurements this definition of hypotension was determined. Subgroup analyses of different MAP thresholds were based on the following classification: MAP < 65–60 mmHg, MAP < 60–55 mmHg, MAP < 55–50 mmHg, MAP < 50–45 mmHg, MAP < 45 mmHg. The combination of severity and duration of hypotension was expressed in TWA. The TWA was quantified by summing the area under the threshold (AUT) (MAP < 65 mmHg), divided by the total duration of the measurement period [3,12]. As described in Figure 1, the units for AUT are mmHg*min and the units for TWA are mmHg. For subgroups classified by different TWA thresholds, we defined three equally sized groups according to number of patients.

Classification of shock was based on the specific duration of BP measurements of the study and determined using the criteria established below, only at the time of all that specific BP measurements. The first step of shock labeling was dividing patients in a shock and non-shock group. Patients were classified to one of the four types of shock or to a combination group, if they suffered from multiple types of shock. Patients with infection, sepsis, sedation or paraplegia were labelled as ‘distributive shock’. Patients with an impaired heart function, inotropic medication or dependent pacemaker rhythm were assigned to the ‘cardiogenic shock’ group. The classification ‘obstructive shock’ was made when tamponade or pulmonary embolism was present, and in case of a hypovolemic situation (i.e., bleeding or dehydration) patients were assigned to the ‘hypovolemic shock’ group. Patients who met multiple criteria, as described above, were placed in the ‘combination shock’ group. Hemodynamic variables of the Flotrac EV1000 monitoring system (cardiac output, contractility, vascular resistance and MAP) and clinical parameters (lactate > 2.0 mmol/L, diuresis < 0.5 mL/kg/hour and the dose of noradrenalin during the measurement (>0.08 mcg/kg/min) were also assessed to support classification into the four shock types. 

### 2.5. Signal Quality and Arterial Waveform Analyses

Before starting the measurements, the transducer of the arterial line system was calibrated (zeroing) and during the measurement a waveform quality check was done at regular time interval. Data were visually checked for completeness and large artefacts present in the raw waveform signal. A custom software algorithm was constructed to detect individual beats and landmarks and subsequently to extract the required variables and determine signal quality (Matlab (version R2018b, The MathWorks, Inc., Natick, MA, USA) (Appendix A). Beat detection was performed according to Zong et al. [13], and signal quality was computed as per the signal quality index of Sun et al. [13,14] After data quality assessment, the entire measurement per patient was used for the analyses of BP thresholds, TWA, number of hypotensive events, duration of hypotensive events and hemodynamic variables.

### 2.6. Data Acquisition

Patient characteristics (age, weight, height, sex, intoxications, medical history, (home) medication, ventilator settings, reason for ICU admission and sequential organ failure assessment (SOFA) score) were all extracted from the electronic patient database (2018, Epic Systems Corporation, Verona, WI, USA). All de-coded data were entered into a GCP compliant database (Castor EDC™, Version 2019.1.5, Amsterdam, The Netherlands) and handled according to the General Data Protection Regulation of May 2018. A 10% check was performed after data entry and less than three percent errors were found which was considered satisfactory.

### 2.7. Statistical Analyses

Data analyses were performed with IBM SPSS® Software (Version 25.0, IBM Corp., Armonk, NY) and R (Version 3.5.1; 2 July 2018, Vienna, Austria). The data set was reviewed for accuracy, missing values, outliers and were screened for normality using histograms, QQ-plots and boxplots. Categorical data were described as frequencies (percentages rounded to whole integers) and analyzed between groups with a Pearson’s Chi-squared (if frequencies were ≤5) or a Fisher’s exact test. Continuous variables were presented as mean with standard deviation (SD) in normally distributed or median with first and third quartile [Q1–Q3]. Group differences were calculated using *t*-tests, Kruskal–Wallis or Mann–Whitney-U for continuous data. Statistical significance was assumed at *p* < 0.05. To determine factors associated with hypotension based on the absolute threshold of 65 mmHg or associated with severe TWA, variables were tested with a univariable analysis. To test for additivity, we created a correlation matrix with the statistically significant and clinically relevant variables. Statistically significant and clinically relevant variables (based on literature and experience) were added to the multivariable regression model and analyzed using a forward-backward stepwise method, with a 1:10 event per variable predictor rate.

## 3. Results

### 3.1. Patient Characteristics

Patient selection is reported in Appendix A. A total of 514 patients meeting the inclusion criteria were screened and 499 patients were included in the data analyses. Table 1 summarizes the patient’s characteristics. The mean age was 61 years (14), most patients were male (327 (66%)), the mean SOFA score was 10 (3) and the most common reason for ICU admission was post-surgery (216 (43%)). Most patients required mechanical ventilation (358 (72%)) and received vasoactive medication (302 (61%)). The median monitoring time per patient was 441 min [411–962 min]. For each patient 98% [95–99%] of available continuous BP data was usable for analyses after evaluation through the signal quality algorithm. The majority of measurements (305 observations (61%)) were obtained during daytime. Of all patients, 293 (59%) were labelled as non-shock and 206 (41%) were diagnosed with shock. Within this last group, 94 (19%) were assigned to the distributive shock group, followed by 66 (13%) to the cardiogenic shock group, 32 (6%) to the combination shock group, 12 (2%) to the hypovolemic shock group and 2 (0.4%) to the obstructive shock group (Appendix A).

### 3.2. Incidence and Severity of Hypotension

Based on the absolute MAP threshold of 65 mmHg, the incidence of hypotension was 75% (376 out of 499) and 9% (46 out of 499) suffered severe hypotension (MAP < 45 mmHg). Clinically relevant and statistically significant differences between patients with and without hypotension were number of day-time measurements (58% (218 out of 376) vs. 71% (87 out of 123); *p* = 0.008), cardiogenic shock (16% (62 out of 376) vs. 3% (4 out of 123); *p* < 0.001), no shock (54% (204 out of 376) vs. 72% (89 out of 123); *p* < 0.001), higher median Apace score (49 [36–63] vs. 34 [32–60], *p* = 0.031) and no treatment with norepinephrine during the measurements (59% (221 out of 376) vs. 45% (55 out of 123); *p* = 0.012). Duration of mechanical ventilation, mortality or length of stay during ICU admission and in hospital admission where comparable between both groups (Table 2).

In patients with hypotension (MAP < 65 mmHg), the median number of events per patient, was one per 73 min (6 events [2–13]/440 min ICU stay [410–926]). The median duration of a single hypotension event was 52 min [5–170]. The median TWA was 0.3 mmHg [0–1.0].

Appendix A summarizes the results of the different subgroups classified according to the severity of hypotension based on different MAP thresholds. Overall, 62% (309 out of 499) of the study population had a MAP below 60 mmHg, 38% (188 out of 499) < 55 mmHg, 20% (99 out of 499) < 50 mmHg and 9% (46 out of 499) < 45 mmHg. Patients in the lowest MAP group were significantly older compared to patients without hypotension (57 years (14) vs. 68 (12); *p* = 0.001, respectively).

### 3.3. Severity Expressed in TWA

Table 3 illustrates the results of the subgroups classified by different TWA thresholds (mild-TWA (range: 0–0.0233 mmHg; *n* = 166), moderate-TWA (range: 0.0234–0.5719 mmHg; *n* = 166) and severe-TWA (range: >0.5720 mmHg; *n* = 167)). Significant overall differences between the mild and severe TWA subgroups were mean age (57 years (14) vs. 62 (15); *p* < 0.001), being classified to cardiogenic shock group (4% (7 out of 166) vs. 22% (36 out of 167); *p* < 0.001), classified to distributive shock group (8% (13 out of 166) vs. 25% (42 out of 167); *p* < 0.001), mean diuresis (1.2 mL/kg/h (1.0) vs. 0.9 (0.8); *p* = 0.002), mean haemoglobin (6.8 mmol/L (1.4) vs. 6.3 (1.2); *p* < 0.001), maximum norepinephrine dose during the measurement period (0.12 mcg/kg/min (0.11) vs. 0.22 (0.20); *p* < 0.001), respectively.

### 3.4. Clinical Factors Associated with Hypotension

Based on multivariable logistic regression analysis reported in Table 4, clinical factors associated with hypotension (MAP < 65 mmHg) were age (OR 1.03 (95% CI 1.01, 1.06; *p* = 0.005)), male sex (OR 2.59 (95% CI 1.38, 4.85; *p* = 0.003)), BMI (OR 0.95 (95% CI 0.90, 1.00; *p* = 0.046)) and cardiogenic shock classification (OR 3.70 (95% CI 1.17, 11.66; *p* = 0.026)) (Appendix A). Only admission to the ICU post cardiac surgery (OR 2.33 (95% CI 1.30, 4.19; *p* = 0.005)) was associated as factor for the development of severe TWA (Appendix A).

## 4. Discussion

In this prospective observational study describing hypotension, in a mixed surgical and non-surgical population in the ICU the main findings are; Firstly, the incidence of hypotension and severe hypotension is high; Secondly, clinical factors associated with hypotension (MAP < 65 mmHg) were age, the male sex, BMI and cardiogenic shock classification; and thirdly, post cardiac surgery was associated with severe TWA. 

### 4.1. Incidence 

The incidence of hypotension found in our prospective study is in line with data from previous retrospective studies [15,16]. We selected MAP thresholds consistent with previously published data and guidelines in order to compare outcomes [2,3,6]. Bighamian et al., reported an incidence of 68% in a reasonably similar population [15]. However, these results were retrospective and based on the definition of MAP below 60 mmHg for more than 15 min. This is in contrast to our definition, where MAP had to be below 65 mmHg for a minimum of ten seconds and with a minimum of one minute in between single hypotensive events based on continuous BP data. In patients with distributive shock using vasopressor support, Vincent et al., found an incidence of hypotension of 62%. Hypotension was defined in this study as an episode of MAP below 65 mmHg with a duration of more than two hours (consecutively). A higher incidence (93.5%) was found for only a single hypotensive event (each single documented MAP < 65 mmHg) [7]. Previous randomized clinical trials, which were performed in patients with septic or vasodilatory shock, have primarily looked at mortality in relation to blood pressure but not at the incidence and severity of hypotensive events [17,18,19].

### 4.2. Severity

Results of previous retrospective studies have shown an association between the duration and severity of hypotension in relation to clinical harm and mortality [3,7]. Calculating TWA, by multiplying the duration of all hypotensive events with the area under the blood pressure threshold is a more accurate method of expressing the cumulative severity of hypotension [3]. Although important for outcome, data on the severity of hypotension expressed in TWA from prospective studies are lacking. The severity of hypotension is associated with poor outcomes, but most studies, retrospective in nature, have only reported the incidence of hypotension or duration of hypotensive events based on absolute blood pressure thresholds [5,9,20]. The severity of hypotension was therefore not accurately determined. Recently, in a retrospective analysis of 8782 septic patients by Maheshwari et al., a strong association was found between each unit mmHg increase in TWA (MAP < 65 mmHg) and mortality, myocardial injury (MI) or acute kidney injury (AKI) [3]. In this study, 37.7% of all patients were found to have a MAP below 55 mmHg [3]. For this specific MAP threshold, the odds ratio for mortality was found to be 1.136 (95% CI [1.095–1.178]; *p* < 0.001), for AKI 1.092 (95% CI [1.063–1.123]; *p* < 0.001) and for MI 1.045 (95% CI [1.004–1.087]; *p* = 0.033). Vincent et al., confirmed these findings and found an even stronger association with mortality as a result of a longer duration in hypotension [7]. Khanna et al., demonstrated a strong association between a lowest median MAP and mortality, MI or AKI in surgical ICU patients [9]. Additionally, in a retrospective cohort study of 111 patients with septic shock, a large difference in duration of hypotension was shown between the 30-day survivors and the group of non-survivors. At a threshold of 65 mmHg, the 30-day survivors spent approximately 6% of their time in hypotension, compared to 40% for the non-survivors. The percentages of time spent in hypotension found in the aforementioned study were not comparable to our findings [5]. In order to compare future studies, we would recommend using TWA as a more uniform read-out that takes both severity and duration into consideration.

### 4.3. Clinical Factors Associated with Hypotension

Patients admitted to the ICU are often hemodynamically and respiratory unstable and frequently have a limited cardiac physiological reserve. Those patients would likely be at increased risk to develop more events of hypotension by a disturbed sympathetic-parasympathetic balance and autonomic dysfunction [21]. Higher age and decreased heart rate variability are indirect measures of autonomous dysregulation [22]. A reduced risk for ischemic events is more profound in women, which may explain why men have a higher risk of developing hypotensive events [22]. Autonomic dysfunction is commonly seen in patients with sepsis, severe brain injuries and myocardial infarction [23]. These data supports our observation that patients after cardiac surgery or patients in cardiogenic shock are at higher risk to develop hypotension during ICU admission.

### 4.4. Limitations

This study has several limitations. We did not investigate the relationship between the severity of hypotension in relation to outcome, but only the occurrence of hypotension in relation to mortality and the duration of ICU admission. Previous studies already demonstrated this relationship. The purpose of the current study was to assess, prospectively, using beat-to-beat data, the incidence, severity and clinical factors associated with hypotension of patients admitted to the ICU.

Secondly, data regarding a patient’s individual cardiovascular physiological reserve are limited. In this study we showed that, based on medical history, age and type of shock in relation to hypotension, individual MAP thresholds can be defined although risk factors were not known. Objective judgement of patient’s individual condition remains difficult. In future studies, data collection with a more personalized approach in the context of individual cardiovascular physiological reserve is worth considering.

Thirdly, this is a single centre observational study, which limits the generalisability of the findings.

### 4.5. Strengths

This study provides insight into the incidence, severity and factors associated with hypotension in a mixed surgical and non-surgical population in the ICU. This study may serve as an important basis for the development of future clinical studies. Waveform analyses, the identification of factors associated with hypotension and the new developments, like machine learning, in the prevention of hypotension, may play a more prominent role in future studies [24,25].

## 5. Conclusions

Hypotensive events are common in a mixed surgical and non-surgical ICU population, where three quarters of all patients experienced an episode of hypotension. Currently, treatment of hypotension is reactive. Whether prevention of hypotension, guided by predictive models incorporating arterial waveform analyses and clinical factors associated with hypotension, will improve outcomes in the ICU setting warrants further research.

## Figures and Tables

**Figure 1 jcm-11-06832-f001:**
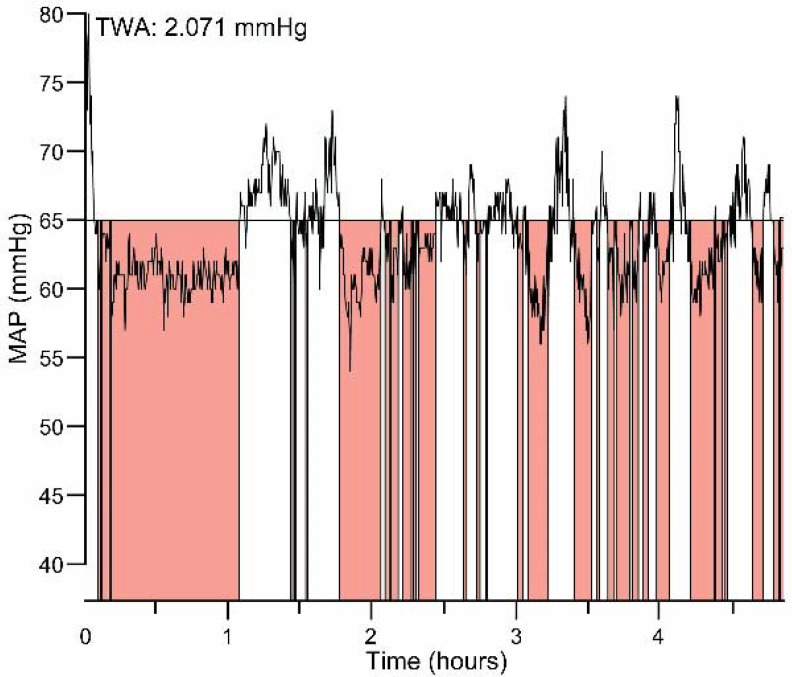
The area under the threshold to calculate the time-weighted average of hypotension. Subscript Figure 1: This is an illustration of a first section measurement (5 h) of one individual patient from this study to indicate how the TWA is calculated.TWA = (depth hypotension below MAP threshold (in this case 65 mmHg) in mmHg × time spent below MAP 65 threshold in minutes. The time-weighted average (TWA) in hypotension combines the time and depth of hypotension. To calculate the TWA of hypotension the AUT is needed. The AUT is calculated as the ‘area of hypotension below the threshold—defined as a Mean Arterial blood Pressure (MAP) of 65 mmHg’ × ‘time spent below MAP 65 mmHg in minutes’. Subsequently the formula of TWA in hypotension is as follows: ‘AUT’/‘total duration of measurement period in minutes’.

**Table 1 jcm-11-06832-t001:** Baseline characteristics.

Baseline Parameters	All Patients *n* = 499
Sex, male, *n* (%)	327 (66)
Age, years, mean (sd)	61 (14)
Number of patients older than 65 years, *n* (%)	221 (44)
Weight (kg), mean (sd)	82.97 (19.5)
Height (cm), mean (sd)	174 (9.9)
BMI, mean (sd)	27 (6)
SOFA score, mean (sd)	10 (3)
Vasoactive medication during measurements, *n* (%)	302 (61)
Mechanical ventilation, *n* (%)	358 (72)
**Measurement details**	
Blood pressure monitoring time per patient (minutes), median [Q1–Q3]	441 [411–962]
Signal quality percentage per patient, median [Q1–Q3]	98.0 [94.6–99.0]
Number of daytime measurements, *n* (%)	305 (61)
Number of night-time measurements, *n* (%)	194 (39)
**Reason of ICU admission**	
Respiratory failure, *n* (%)	57 (11)
Neurological disease, *n* (%)	82 (16)
Sepsis, *n* (%)	38 (8)
Cardiac shock/other cardiac, *n* (%)	19 (4)
Postoperative after surgery, *n* (%)	216 (43)
**Assigned shock groups**	
Cardiogenic shock, *n* (%)	66 (13)
Distributive shock, *n* (%)	94 (19)
Hypovolemic shock, *n* (%)	12 (2)
Obstructive shock, *n* (%)	2 (0.4)
Combination type of shock, *n* (%)	32 (6)
Non-shock classification, *n* (%)	293 (59)

Statistic presented as mean (standard deviation), median [first quartile, third quartile] or number of patients (%). Abbreviations: MAP, mean arterial pressure; BMI, body mass index; SOFA, sequential organ failure assessment.

**Table 2 jcm-11-06832-t002:** Hypotension (mean arterial pressure < 65 mmHg).

	Hypotension Classification		
Baseline Parameters	Absent MAP ≥ 65 mmHg *n* = 123 (25%)	Present MAP < 65 mmHg *n* = 376 (75%)	*p*-Value
Sex, male, *n* (%)	71 (58)	256 (68)	0.062
Age, years, mean (sd)	58 (14)	62 (14)	**0.003**
BMI, mean (sd)	28 (6)	27 (6)	**0.049**
Day-time measurements, *n* (%)	87 (71)	218 (58)	**0.008**
**Medical history**			
Myocardial infarction, *n* (%)	11 (9)	59 (16)	0.251
Hypertension, *n* (%)	37 (30)	109 (29)	0.819
Pulmonary disease, *n* (%)	1 (1)	12 (3)	0.314
Diabetes mellitus type II, *n* (%)	15 (12)	66 (18)	0.204
Cerebral vascular accident, *n* (%)	6 (5)	18 (5)	1.000
Gastrointestinal disease, *n* (%)	19 (16)	57 (15)	0.427
Renal insufficiencies, *n* (%)	6 (5)	29 (8)	0.664
Oncological disease, *n* (%)	7 (6)	16 (4)	0.293
**Reason of ICU admission**			
OHCA, *n* (%)	4 (3)	27 (7)	0.136
Post cardiac surgery, *n* (%)	32 (26)	167 (44)	**<0.001**
Intracranial bleeding (SAB), *n* (%)	27 (22)	24 (6)	**<0.001**
Sepsis, *n* (%)	12 (10)	26 (7)	0.326
**Assigned shock groups**			
Cardiogenic shock, *n* (%)	4 (3)	62 (16)	**<0.001**
Distributive shock, *n* (%)	20 (16)	74 (20)	0.506
Hypovolemic shock, *n* (%)	1 (1)	11 (3)	0.309
Obstructive shock, *n* (%)	1 (1)	1 (0.3)	0.430
Non-shock classification, *n* (%)	89 (72)	204 (54)	**<0.001**
**Clinical data**			
Length of stay ICU (days), median [Q1–Q3] *	3 [1–10]	2 [1–7]	0.407
Length of stay in hospital (days), median [Q1–Q3] *	10 [5–23]	10 [6–20]	0.877
Lactate (mmol/L), median [Q1–Q3]	1.5 [1.2–2.0]	1.4 [1.1–2.0]	0.985
Diuresis (ml/kg/h), median [Q1–Q3]	0.93 [0.61–1.48]	0.87 [0.58–1.32]	0.377
Haemoglobin (mmol/L), median [Q1–Q3]	6.8 [5.9–7.8]	6.5 [5.7–7.4]	0.118
Saturatio*n* (%), median [Q1–Q3]	94 [91–96]	93 [71–96]	**0.002**
No treatment with Norepinephrine, *n* (%)	55 (45)	221 (59)	**0.012**
Propofol sedation, *n* (%)	39 (32)	142 (38)	0.280
Mechanical ventilation, *n* (%)	83 (68)	274 (73)	0.356
Days of mechanical ventilation during admission ICU, median [Q1–Q3] *	12 [2–74]	8 [2–47]	0.299
Apache score, median [Q1–Q3] *	34 [32–60]	49 [36–63]	**0.031**
Died during ICU admission, *n* (%) *	12 (9.8)	54 (14.4)	0.220
Died during hospital admission, *n* (%) *	18 (14.6)	70 (18.6)	0.340
**Haemodynamic data**			
Systolic BP (mmHg), median [Q1–Q3]	133 [114–163]	121 [108–143]	**0.005**
Diastolic BP (mmHg), median [Q1–Q3]	60 [52–69]	55 [50–62]	**<0.001**
Number of events per patients, median [Q1–Q3]	-	6 [2–13]	
Total duration of events per patient (min), median [Q1–Q3]	-	52 [5–170]	
Total percentage duration of measurement in hypotensio*n* (%), median [Q1–Q3]	-	9.3 [0.7–29.1]	
TWA per patient (mmHg), 10 s median [Q1–Q3]	-	0.3 [0.03–1.0]	

Abbreviations: BMI, body mass index; BP, blood pressure; MAP, mean arterial pressure; OHCA, out of hospital cardiac arrest; TWA, time weighted average. Statistic presented as mean (standard deviation), median [first quartile, third quartile] or number of patients (%). Group differences were tested with Kruskal–Wallis, Mann–Whitney U or Chi square test. * Missing data of EPD of 38 patients.

**Table 3 jcm-11-06832-t003:** Subgroups of hypotension based on time-weighted average.

	Mild TWA Group (≤0.0233 mmHg) *n* = 166 (33%)	Moderate TWA Group (0.0234–0.5719 mmHg) *n* = 166 (33%)	Severe TWA Group (≥0.5720 mmHg) *n* = 167 (34%)	*p*-Value
**Baseline parameters**				
Sex, male, *n* (%)	96 (58)	117 (71)	114 (68)	**0.035** ** ^a^ **
Age, years, mean (sd)	57 (14) ^¥®^	63 (12) ^¥^	62 (15) ®	**<0.001**
BMI, mean (sd)	28 (7)	27 (6)	27 (6)	0.422
SOFA score, median [Q1–Q3]	9 [6–11] ^¥®^	11 [9–12] ^¥^	10 [9–12] ®	**<0.001**
Temperature °C, mean (sd)	37.0 (0.8) ^¥®^	36.7 (0.8) ^¥^	36.7 (0.8) ®	**0.006**
**Medical history**				
Myocardial infarction, *n* (%)	17 (10)	25 (15)	28 (17)	0.731
Hypertension, *n* (%)	52 (31)	43 (26)	51 (31)	0.502
Diabetes mellitus type 2, *n* (%)	27 (16)	23 (14)	31 (19)	0.507
Cerebral vascular accident, *n* (%)	8 (5)	7 (4)	9 (5)	0.883
Renal disease and insufficiencies, *n* (%)	14 (8)	7 (4)	14 (8)	0.097
**Reason of ICU admission**				
OHCA, *n* (%)	8 (5)	9 (5)	14 (8)	0.353
IHCA, *n* (%)	2 (1)	-	7 (4)	**0.013**
Post cardiac surgery, *n* (%)	38 (23)	77 (46)	84 (50)	**<0.001**
Pneumonia, *n* (%)	2 (1)	11 (7)	5 (3)	**0.026**
Intracranial bleeding (SAB), *n* (%)	42 (25)	5 (3)	4 (2)	**<0.001**
Neurological other, *n* (%)	19 (11)	6 (4)	3 (2)	**<0.001**
Sepsis, *n* (%)	9 (5)	13 (8)	16 (10)	0.356
Assigned shock groups				
Cardiogenic shock, *n* (%)	7 (4)	23 (14)	36 (22)	**<0.001**
Distributive shock, *n* (%)	13 (8)	39 (24)	42 (25)	**<0.001**
Hypovolemic shock, *n* (%)	2 (1)	5 (3)	5 (3)	0.466
Obstructive shock, *n* (%)	-	1 (1)	1 (1)	0.603
Combination type of shock, *n* (%)	2 (1)	13 (8)	17 (10)	**0.002**
Non-shock classification, *n* (%)	141 (85)	85 (51)	67 (40)	**<0.001**
**Clinical data**				
Lactate (mmol/L), mean (sd)	1.6 (1.2)	1.7 (1.5)	2.07 (1.9)	0.088
Diuresis (ml/kg/h), mean (sd)	1.2 (1.0) ^®^	1.0(0.79)	0.9 (0.8) ^®^	**0.002**
Saturatio*n* (%), median [Q1–Q3]	93 [89–95] ^¥®^	92 [70–94] ^¥^	92 [68–94] ^®^	**<0.001**
Haemoglobin (mmol/L), mean (sd)	6.8 (1.4) ^¥®^	6.5 (1.2) ^¥^	6.3 (1.2) ^®^	**<0.001**
Maximum need for norepinephrine dose (mcg/kg/min) during measurement), mean (sd)	0.12 (0.11) ^®^	0.16 (0.13) ^‡^	0.22 (0.20) ^®‡^	**<0.001**
Minimum need for norepinephrine dose (mcg/kg/min) during measurement, mean (sd)	0.07 (0.13)	0.05 (0.08) ^‡^	0.11 (0.15) ^‡^	**0.002**
Mechanical ventilation, *n* (%)	108 (65)	124 (75)	125 (75)	0.077
**Haemodynamic data ***				
Cardiac output (L/min), median [Q1–Q3]	6.0 [4.9–7.5] ^¥^	5.6 [4.5–6.9] ^¥^	5.6 [4.7–6.7]	**0.027**
Systolic BP (mmHg), median [Q1–Q3]	140 [124–158] ^¥®^	117 [107–128] ^¥^	109 [99–120] ^®^	**<0.001**
Diastolic BP (mmHg), median [Q1–Q3]	67 [61–72] ^¥®^	57 [54–62] ¥‡	51 [48–55] ^‡®^	**<0.001**
MAP (mmHg), median [Q1–Q3]	89 [82–101] ^®^	75 [73–80] ^‡^	69 [66–71] ^®‡^	**<0.001**

Abbreviations: BMI, body mass index; BP, blood pressure; IHCA, in hospital cardiac arrest; MAP, mean arterial pressure; OHCA, out of hospital cardiac arrest; SOFA, sequential organ failure assessment; TWA, time weighted average. Statistic presented as mean (standard deviation), median [first quartile, third quartile] or number of patients (%). Group differences were tested with Kruskal–Wallis, One-way ANOVA or Chi square test/Fisher Exact. Statistical significance. * Haemodynamic data (median of measurement period). Asymptotic post hoc significances (2-sided tests) are displayed with: ^¥^ significant difference mild TWA vs. moderate TWA. ^®^ significant difference mild TWA vs. severe TWA. * ^‡^ significant difference moderate TWA vs. severe TWA. ^a^ post hoc analysis non significant after bonferroni correction.

**Table 4 jcm-11-06832-t004:** Multivariable logistic regression analysis.

	Hypotension MAP < 65 mmHg				Severe TWA Group TWA > 0.5720 mmHg				
Covariate	Odds ratio	95% C.I.		*p*-Value	Covariate	Odds ratio	95% C.I.		*p*-Value
		**Lower**	**Upper**				**Lower**	**Upper**	
Age	1.03	1.01	1.06	**0.005**	Reason of ICU admission: post cardiac surgery	2.33	1.30	4.19	**0.005**
Sex (Male)	2.59	1.38	4.85	**0.003**	Reason of ICU admission: IHCA	5.99	0.63	56.74	0.119
BMI	0.95	0.90	1.00	**0.046**	Reason of ICU admission: Intracranial bleeding (SAB)	0.14	0.01	1.45	0.100
Cardiogenic shock classification	3.70	1.17	11.66	**0.026**	Non-shock classification	0.59	0.31	1.12	0.104
Non-shock classification	0.55	0.27	1.14	0.106	Haemoglobin (mmol/L) during measurement	0.84	0.68	1.04	0.108
Minimum need for norepinephrine dose (mcg/kg/min) during measurement	0.12	0.01	1.23	0.074	Maximum need for norepinephrine dose (mcg/kg/min) during measurement	7.14	0.83	61.42	0.073
					Minimum need for norepinephrine dose (mcg/kg/min) during measurement	31.80	0.95	1061.59	0.053

## Data Availability

The datasets used and/or analysed during the current study are available from the corresponding author on reasonable request.

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
