# Peer review of "Incidence, Severity and Clinical Factors Associated with Hypotension in Patients Admitted to an Intensive Care Unit: A Prospective Observational Study"

_jcm, 2022, doi:10.3390/jcm11226832_

Round 1

Reviewer 1 Report

1. Abstract: This is a cohort study and not a trial. Please correct. 

2. Round up the ORs to two decimal points. 

3. Mention the limitation that this is a single centre study which limits the generalisability of the findings. 

4. Was there a risk of participation/selection bias?

Reviewer 2 Report

Your work analyzed the occurrence of low blood pressure (mean arterial pressure <65mmHg) in an ICU population of a mixed type, using information extracted from the direct intravascular blood pressure waveform through an algorithm to compute the intensity of the hypotension and time exposure. 

If I understood correctly, you have used data from 8 hours of wave recording. Probably because of the data bank dimension, but I would like some information regarding the choice for 8 hours and the choice for 10s as a minimum.

Although the two main new concepts presented are explicit, there's a problem with the need to incorporate them into clinical reasoning. Any thoughts on that?

Your work is observational and descriptive and, as you mentioned in the "limitations," did not look at the outcomes; nevertheless, I consider it very relevant, pointing to new uses of monitoring data.

Please check figure 1 legend

Definig new operative variables to be used that can 

Reviewer 3 Report

Patients addmitted to the ICU of one hospital were recruited for this study, in which the condition of hypotension was more finely defined.  Unfortunately, no data can be found in the manuscript.  The study is incomplete because it describes only  the analysis without showing what the results mean in terms of patient care and prognosis: the study is "50% done".  

Round 2

Reviewer 3 Report

Because this is a prospective study, what is presented is reasonable, and I apologize for my comment (i.e.  50% done) made in the first review.

Now that the paper is with figures and tables, I can offer furhter comments.

1.  Title.  ".... in the intensive care unit..."  This is an overstatement of the study.  The study was done in one ICU unit, which could have some unique patient populations, age groups, disease types, race, etc.  The data may not be universaly applicable.  I suggest using "an" instead of "the".

2.  ICU patients are special patients who need constant, all sorts of  interventions.  Looking at Fig. 1, this patient was monitored for almost 5 hours, and many hypotension events were recorded.  Was this patient (and all other patients for that matter) left alone (without treatment), or was (s)he treated?  If the latter, when, how, for how long, etc.  Undoubtedly, these intervantions could change the patient's blood hemodynamics.  In the data analysis, how did the authors deal with various effects of interventions?  The flipside of this is patients may no longer have hypotension if treated well, but this may not mean whatever the type of hypotension these patient had initially causes further issues.

3.  The conclusion of the study is, "In a mixed surgical and non-surgical ICU population, the incidence of hypotension is remarkably high."  Is this the hypothesis you tried to test in this study?  I am sorry, but this conclusion is rather unremarkable.  Could it be that the study is "favoring" data from  patients who are more heavily affected?

4.  Hypotension must be treated immediately, and this may be the reason why a single threshold value is used in practice.  TWA is clearly more accurate.  However, if it takes time to calculate this, patients may not last long enough to be treated.  How useful is TWA as an ICU diagnostic strategy?
